# The relative influence of agricultural abandonment and semi-natural habitats on parasitoid diversity and community composition

**Marina Mazón[1], Santiago Bordera[1], Alexander Rodríguez-Berrío[2], Enric Frago [3]***

**1** Departamento de Ciencias Ambientales y Recursos Naturales, Universidad de Alicante, San Vicente del Raspeig, Alicante, Spain, **2** Departamento Académico de Entomología, Facultad de Agronomía, Universidad Nacional Agraria La Molina, Distrito La Molina, Lima, Perú, **3** CIRAD, UMR CBGP, INRAE, Institut Agro, IRD, Université Montpellier, Montpellier, France

* enric.frago@cirad.fr

**Data Availability Statement:** The data used in this study and the R code used in the analyses can be found in the following public repository: https://zenodo.org/doi/10.5281/zenodo.10568982.

## Abstract

Wild biodiversity is usually larger in semi-natural habitats than in croplands, but this pattern is not ubiquitous because it varies among taxa and geographic regions. Knowing how the diversity of natural enemies is structured at the landscape level is important to better understand when semi-natural habitats promote the conservation of natural enemies and ultimately enhance biocontrol. We explore the relative influence of agricultural abandonment and the proportion of semi-natural habitats at the landscape level on the diversity and abundance of parasitoid wasps in the Ichneumonidae family. We studied changes in parasitoid diversity both at local and regional scales (i.e. alpha vs beta diversity), and both at the taxonomic and functional level (i.e. species vs guild identities). We extracted landscape features in circular buffers of varying radii to perform a multi-scale analysis, and to assess at which scale landscape-level effects influenced parasitoid assemblages. We found that parasitoid alpha and beta diversity decreased with an increasing proportion of semi-natural habitats. The multi-scale analysis revealed that for this group of natural enemies, landscape-level effects occur at mid to low distances (i.e. less than 500m). Our results provide insights into the origin of pest natural enemies, their spillover to croplands, and may help to understand under which circumstances semi-natural habitats fail at promoting biocontrol services.

## 1. Introduction

More than a third of the current global terrestrial land is dedicated to agriculture, 30% to forestry, 25% to human settlements, and only the remaining 10% is occupied by intact or primary forests [1, 2]. Agriculture has a strong impact on natural communities as it often implies ploughing, and a reduction in structural as well as habitat complexity, with long-lasting negative effects on plant diversity and productivity [3], soil organic carbon, primary consumers, and the structure of trophic webs [4–7]. Human activities have thus led to an unprecedented

**Funding:** This study was supported by the Projects BOS2000-0148 from Ministerio de Ciencia y Tecnología (D.G.I.) of Spanish Government and GV06/271 from Conselleria d'Empresa, Universtitat i Ciència of Generalitat Valenciana (Spain). AR-B was financed by a PhD research grant MAE-AECI (2003-2006) of Agencia Española de Cooperación Internacional (Spanish Goverment). EF is currently funded by the Agence Nationale de la Recherche (ANR) via the ANR ENEMYCOCKTAIL project and by CIRAD. The funders had no role in study design, data collection and analysis, decision to publish, or preparation of the manuscript.

**Competing interests:** The authors have declared that no competing interests exist.

loss and alteration of natural habitats, and has restricted them to small patches interspersed within human settlements, mono-cultural croplands, pasturelands and semi-natural habitats. Semi-natural habitats are abundant in Europe, and are dominated by natural communities with a varying degree of influence of human activities. A common type of semi-natural habitat is that derived from abandoned agricultural lands. These lands have become an important element of the landscape in Europe, particularly due to the set-aside incentive schemes that promote leaving a part of the agricultural land out of intensive production [8]. Set-aside incentives can promote planting specific cover crops, establishing buffer strips or preserving particular habitats for wildlife, but in many cases abandoned croplands simply follow a secondary succession. In this later case, croplands are recolonised by natural vegetation and fauna, where the influence of agriculture slowly fades away [9].

The presence of semi-natural habitats near croplands may determine crop health because they may provide refuges to wild biodiversity and ensure ecosystem services like pollination, biocontrol or nutrient cycling [10–13]. Among these services, biocontrol is the most intensely studied. It is often accepted that there is higher pest suppression in croplands that are embedded within landscapes dominated by semi-natural habitats because these habitats usually harbour diverse assemblages of natural enemies [6, 12, 14–17] that ultimately favour higher-order control in food-webs [18]. The mechanisms by which semi-natural habitats promote natural enemy diversity remain elusive and have rarely been demonstrated experimentally, but they likely include cascading effects of primary productivity, plant diversity, and plant biomass. Relative to croplands, all these variables are often larger in semi-natural habitats [3, 15–17]. The paradigm that semi-natural habitats enhance enemy diversity and pest suppression, however, has often been challenged [19, 20], particularly by a recent meta-analysis that showed strong variation across studies and systems on the response of insect natural enemies to landscape composition [19]. A simple reason why semi-natural habitats may not enhance pest suppression in nearby croplands is because natural enemies may be more abundant and diverse in croplands (e.g. [21, 22]). Even if semi-natural habitats have diverse communities of natural enemies, these habitats may not improve biocontrol services if enemies engage in antagonistic interactions among them [e.g. 23] or if they do not spillover to croplands. Natural enemy spillover among semi-natural habitats and croplands is a question that has received little attention, even if it is a crucial element to understand natural biocontrol services. Spillover can be measured directly as done by Inclán and coauthors (2016) [22] who measured it using pan-traps that intercepted enemies moving from crop margins to croplands. Spillover can be indirectly measured by comparing the composition of natural enemy assemblages (i.e. beta diversity) between semi-natural habitats and croplands as done in [24]. More specifically, beta diversity can be partitioned into species replacement and richness differences [25–27]. When replacement dominates, some species are lost from sites and others take place indicating little spillover among habitats. When richness differences dominate, however, some sites have more unique species, which suggests that enemy assemblages in poor habitats are acquired via spillover from more diverse ones. In the particular case of croplands and semi-natural habitats, if both habitats contain independent natural enemy assemblages (i.e. when replacement dominates), it is unlikely that one habitat influences the other. Beta diversity indices are increasingly recognized as important for biodiversity conservation [28], and emerging evidence suggests that these indexes can provide unique insights into biodiversity patterns [29–31]. Surprisingly, however, the use of these methods to understand insect dispersal between habitats is rarely used.

Another important element to understand the relationship between biodiversity and biocontrol is to focus on natural enemies' functional roles [32], particularly because one of the main mechanisms by which natural enemy diversity and biocontrol may relate positively is

through complementarity [15–17, 33]. Complementarity implies that natural enemies differ at the ecological (or functional) level, thus partitioning their ecological niches, for example by attacking different host species or hosts found in different habitats. Natural enemy functional diversity usually favours pest biocontrol [34] because even if natural enemies are taxonomically diverse, they may fail at suppressing herbivores if they perform very similar functions at the ecological level [15, 35]. Diversity has many different facets that can be measured as taxonomic, functional and phylogenetic, each measure being complementary to others (e.g. [36]). Taxonomic diversity has so far been used extensively to explore the impact of agriculture on animal communities, but more studies using functional diversity indexes in this context are needed. This can be particularly useful for the study of natural enemies because functional traits are often linked to prey use and therefore to the type and strength of the biocontrol services that different communities may provide [35, 37, 38].

In this work we studied the abundance, diversity and community composition of hymenopteran parasitoids belonging to the family Ichneumonidae in a landscape composed of natural, semi-natural and agricultural habitats. Hymenopteran parasitoids, and ichneumonids in particular, are a very diverse group of natural enemies, and are usually susceptible to any changes occurring at lower levels in the food chain [39, 40]. For these reasons, Ichneumonidae are an important group of pest natural enemies, and have often been used as bioindicators of arthropod biodiversity and ecosystem disturbances [41–44]. We sampled parasitoids in replicated sites in a Mediterranean ecosystem, and we classified them into parasitoid guilds (following [43]) so that we explored diversity patterns at both the taxonomic and at the functional level. We explored the relative contribution of two different layers to alpha and beta diversity. At the local level, the agricultural history of the site was represented by the time since agriculture was abandoned. At the landscape-level, the effects of current agricultural practices were represented by the relative proportion of forests and semi-natural habitats in the area surrounding each study site. To estimate the spatial scale at which these effects may operate we also performed a multi-scale analysis [45, 46]. Even if there are many examples where semi-natural habitats do not enhance biocontrol services [19], we stick here to the most accepted paradigm that semi-natural habitats increase natural enemy diversity. We therefore test the following hypotheses (Fig 1). (1) We hypothesise that alpha diversity (i.e. species richness and evenness) and insect abundance will increase with the time since agriculture was abandoned and with the proportion of semi-natural habitats at the landscape level (i.e. the proportion of semi-natural habitats in buffers surrounding study sites). (2) We hypothesise that beta diversity will increase with the time since agriculture was abandoned and with the proportion of semi-natural habitats at the landscape level. We also partition beta diversity into species replacement and richness differences to better understand parasitoid spillover among habitats [25–27]. (3) We also hypothesise that functional richness and evenness (i.e. the number and diversity of parasitoid functional guilds) will relate positively with the time since agricultural abandonment, and with the proportion of semi-natural habitats at the landscape level [47, 48].

## 2. Material and methods

### 2.1. Study area

We carried out this study in two mountain ranges in the Valencian region of the South-eastern Iberian Peninsula: Carrasqueta and Mariola. These mountains belong to two nearby provinces, Alicante and València, respectively. Both mountains have a common biogeographic history, have a West to East orientation and are separated by a wide valley of approximately 15 km. The climate is Mediterranean with annual mean temperatures between 13 and 16°C, and accumulated annual rainfall of around 500 mm. In this area the landscape is a mosaic of natural

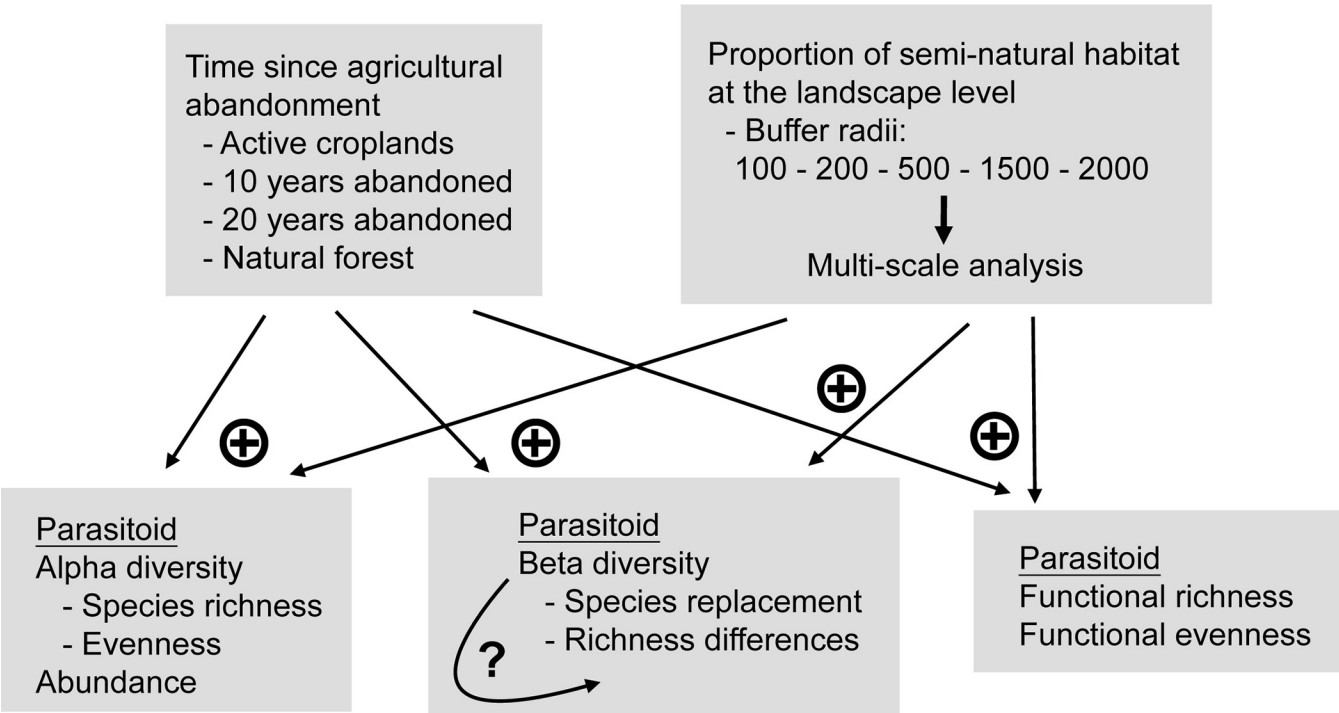

**Fig 1. Conceptual diagram of the experimental design and variables tested in this study.** Arrows point from predictor to response variables. Positive and negative symbols represent our a priori hypotheses.

protected areas characterised by native woodlands of *Quercus ilex* subsp. *rotundifolia* (Lam.) Shwartz ex T. Morais and *Q. coccifera* L, embedded within small-scale croplands. These croplands comprise mostly dry-land wheat cereal plantations, many of which are abandoned. In each mountain range, we selected four sites along a gradient of time since agriculture was abandoned, leading to a total of eight study sites. At each mountain range we studied one site located in an active agricultural cropland, two sites in which agriculture was abandoned after ten or twenty years, respectively, and one site in a wild protected area (Table 1). These sites

**Table 1. Description of sites where the study was carried out.**

| | Habitat type | Habitat vegetation | Altitude (masl) | UTM coordinates |
|---|---|---|---|---|
| **Mariola mountain** | | | | |
| Caveta del Buitre | wild area | Mediterranean shrubland with pines and some holm oaks | 1,200 | 30SYH161937 |
| Retura | 20 years ago abandoned crops | Shrubland with abandoned olive trees in terraces | 900 | 30SYH160943 |
| Foia Ampla | 10 years ago abandoned crops | Cereal pastureland in terraces, with some herbaceous communities | 1,060 | 30SYH169931 |
| Mas del Parral | managed crop lands | Wheat crops with natural grasslands | 900 | 30SYH142931 |
| **Carrasqueta mountain** | | | | |
| Menejador | wild area | Mediterranean shrubland with holm oaks and some pines | 1,352 | 30SYH143819 |
| Venta Carrasqueta | 20 years ago abandoned crops | Shrubland with abandoned crops in terraces | 980 | 30SYH191770 |
| Mas de San Ignacio | 10 years ago abandoned crops | Cereal pastureland in terraces, with Mediterranean shrubland and some scattered pines | 1,020 | 30SYH187766 |
| Mas de Cano | managed crop lands | Wheat and sunflower crops with natural grasslands | 940 | 30SYH201775 |

also varied at the landscape level in terms of the proportion of natural and semi-natural habitats surrounding them (i.e. from 60 to 100%). The sites were selected to avoid collinearity between the time since sites were abandoned from agriculture, and the proportion of natural or semi-natural landscapes.

## 2.2. Sampling and determination

We sampled the insects during the warm months of the year, from May to November in 2001 and 2002. To collect the insects, we placed a Townes style Malaise trap in each site. We filled collecting pots with ethanol 70% and we replaced them every 15 days. We mounted and determined to at least, the subfamily level all Ichneumonidae specimens captured using the keys from [49]. We identified most subfamilies to species level, with the exceptions of Ichneumoninae, Mesochorinae, Phygadeuontinae, Campopleginae and Ctenopelmatinae. All insects are currently stored at the Entomological Collection of Alicante University (CEUA). Insects were collected from eight sites (four habitat types replicated in two mountain ranges, see more details below) over two consecutive years. Each site was sampled 10 times per year in 2001 (from the 1st of May to the 30th of September) and 13 times in 2002 (from the 1st of May to the 15th of November), and yearly samples belonging to the same site were pooled together and considered as a single replicate.

## 2.3. Trophic guilds

We classified the parasitoids into trophic guilds according to host range and the feeding niche of the host following [43]. Trophic guilds represent host use by parasitoids and are therefore a reliable representation of their functional role in the ecosystem. Mazón and Bordera (2014) [43] recognized nine parasitoid categories: parasitoids of exposed grazing phytophagous larvae (gPh), parasitoids of weakly concealed phytophagous larvae (cPh), parasitoids of cocoons and pupae (Coc), parasitoids of fungus-feeding larvae (Fun), parasitoids of saprophagous larvae (Sap), parasitoids of zoophagous and predatory larvae (Zoo), parasitoids of xylophagous larvae (Xyl), parasitoids of melitophagous larvae (i.e. larvae that feed on honey or nectar) (Mel) and polyphagous parasitoids (Pol). We excluded from functional analyses those parasitoids that could not be associated to any category.

## 2.4. Estimating landscape features and the multi-scale analysis

We obtained the proportion of agricultural relative to natural or semi-natural habitat from five nested circular buffers surrounding study sites. This relative measure is a common landscape metric used in the study of natural enemy effects on herbivores [19]. These buffers were of 100m, 200m, 500m, 1000m, 1500m, and 2000m radii. We extracted data from a vector layer obtained from the CORINE Land Cover project, which was updated in 2000 (https://land.copernicus.eu/pan-european/corine-land-cover). This vector layer was the most commonly used when insects were sampled. We extracted information from CORINE layers with *QGIS* Desktop version 2.18.2, which was subsequently exported to R. The vector layer used contained nine habitat types, four considered as natural or semi-natural vegetation and that included conifer forests, shrubland, mixed forest, and sclerophyllous forest. These habitats were considered as semi-natural because even if some forests are protected they are not pristine habitats as they have a certain degree of influence from past human activities. Agricultural lands included mixed agricultural lands, rainfed cultures, orchards, olive groves, and agricultural land containing natural vegetation.

Since we had little information on dispersal capabilities of the parasitoids studied, and of the scale that landscape features may affect them, we performed a multi-scale analysis

following the method proposed by Fahrig (2013) [45] and Jackson and Fahrig (2015) [46]. This method is based on the principle that the relationship between landscape features and any population or community-level measure should be stronger at the scale that best fits the species studied. The strength of the relationship is measured in statistical methods with goodness of fit metrics, commonly with R-squared or pseudo R-squared values [45, 46, 50]. We obtained the proportion of semi-natural habitats surrounding study sites using five nested circular buffers centred at each study site: 100m, 200m, 500m, 1000m, 1500m, and 2000m. These radii likely represent short and long-distance scale effects. It is expected that the deviance explained by the landscape features studied (i.e. the proportion of semi-natural habitats) in the models will steadily increase with buffer size until it peaks at the scale that best fits the dispersal abilities of the studied species. After the peak, explained deviance is expected to decrease along with the dilution of the scale effect. We built for each of the different response variables (i.e. parasitoid diversity and abundance) and buffer radii an independent mixed effects model including both time since agricultural abandonment and the proportion of semi-natural habitats at the landscape level. This led to a total of 24 models (model details are provided below). We assessed the deviance explained by landscape effects by comparing the deviance explained by full models, relative to that of models excluding the landscape-related variable with the *nagelkerke* function of the *rcompanion* package [51]. We tested the significance of landscape effects by comparing simplified and complex models, and within the same response variable we corrected p-values for multiple testing using the false discovery rate method. This method is appropriate when a large number of comparisons are performed [52]. Model assumptions were checked as explained in the following section for other mixed effects models.

In our study some sites were located a few kilometres apart. To ensure the lack of spatial covariation in landscape-extracted variables we checked the correlation between landscape variables and kilometric distances between sites. For each buffer radii we built a matrix representing pairwise differences in the proportion of semi-natural habitat between sites. We also built a matrix of pairwise kilometric distances between study sites. The kilometric distances matrix was compared against each of the different landscape-level matrices obtained from each buffer radii using the Mantel test run for $10e^4$ permutations in the *vegan* package in R [53].

## 2.5. Diversity estimates, beta diversity partitioning and mixed effects models

We performed all analyses in R version 4.1.2 [54]. At the taxonomic level, abundance was estimated as the total number of insects sampled, and alpha diversity (i.e. the diversity inherent to each ecosystem) as species richness and evenness. To account for potential undetected species, alpha diversity was estimated using abundance-based rarefaction and extrapolation methods [55, 56], that were further extended to Hill numbers (or effective number of species) [57]. Species richness (taxonomic richness) and the Simpson diversity index (taxonomic evenness) were thus estimated as Hill numbers of order $q = 0$ and $q = 2$, respectively, using the *iNEXT* function in the *iNEXT* package in R [58]. The number of different guilds present in each community (functional group richness), and the Simpson index based on the accumulated number of individuals in each parasitoid trophic guild (functional group evenness) were also estimated using abundance-based rarefaction and extrapolation methods [59]. Regarding beta diversity (i.e. the differences between parasitoid assemblages occurring in each ecosystem), differences in species composition among habitat types were explored with the local contribution to beta diversity (LCBD) index. This metric estimates how each site contributes to global beta diversity and represents the uniqueness of each site [60]. The LCBD metric was obtained with the function *beta.div* in the *adespatial* package [61]. This same package was used to partition beta

diversity into species replacement and richness differences with the function *beta.div.comp*. When replacement dominates, some species are lost from sites and others take place, whereas when richness differences dominate, some sites have more unique species than others [25–27]. The function *beta.div* also provided a *p*-value (obtained via 999 permutations) for each LCBD value obtained, which tests their individual significance and contribution to beta diversity.

To test the relative contribution of agricultural abandonment and of agricultural practices at the landscape level to insect diversity and abundance, we built an independent mixed effects model with a Gaussian distribution for each of the following response variables: species richness, species evenness, insect abundance and beta diversity. Each model included as predictors time since agricultural abandonment and the proportion of semi-natural habitats surrounding each site. To account for the non-independence of samples collected on the same site but on different years, we built a random slopes model that included as a random factor year within time since abandonment, and year within the proportion of semi-natural habitat. This design ensures that data obtained each year is uniquely identified within each explanatory variable [62]. Linearity and normality of model residuals, the absence of collinearity between explanatory variables and the absence of highly influential observations (i.e. Cook's distances) were visually assessed using the *check_model* function in the *performance* package [63]. To satisfy model assumptions, species richness, the Simpson diversity index, and insect abundance were standardised to mean and standard deviation equal to zero and one, respectively [64]. In addition, the *lmerControl* function with the optimiser *nloptwrap* from the package *nloptr* was used to improve model performance [65]. As a measure of Goodness-of-Fit the variance explained by the models was obtained with the function r.squaredGLMM from the package MuMIn [66]. Numeric values related to time since agriculture was abandoned were given to the different habitat types: zero for active croplands, ten and twenty for croplands abandoned for ten and twenty years, respectively, and 75 to intact forests. Although intact forests have never been cleared for agricultural use, the 75 value represents a time frame that ensures that forests were not altered by human activities like extraction of oak wood for charcoal production [67]. The variable considering time since agriculture was abandoned was transformed into continuous to allow for an easy correlation with diversity estimates and to prevent the loss of degrees of freedom when using it as categorical. The Spearman's correlation test was used to explore the collinearity between the two main variables studied here.

## 3. Results

In this study, 2,721 Ichneumonids were collected. These specimens were classified into 161 species (S1 Table), belonging to the following 15 subfamilies: Acaenitinae, Anomaloninae, Banchinae, Collyriinae, Cremastinae, Cryptinae, Diplazontinae, Metopiinae, Ophioninae, Orthocentrinae, Orthopelmatinae, Pimplinae, Tersilochinae, Tryphoninae and Xoridinae. The subfamily Cryptinae represented 75% of the total number of specimens determined. Among them, we sampled 216 specimens of *Trychosis legator* (Thunberg 1822), which was the most abundant species in sites with active croplands, or in those abandoned for ten years. A total of 230 specimens of *Mesostenus albinotatus* Gravenhorst, 1829 were also collected, the species being the most abundant in sites where croplands were abandoned for 20 years. *Collyria distincta* Izquierdo & Rey del Castillo, 1985 was the most abundant species in semi-natural habitats with a total of 155 specimens sampled.

### 3.1. Scale analysis

For the different diversity measures the deviance explained by landscape effects was larger when extracting information from buffers of radii smaller than 1000 metres, except for the

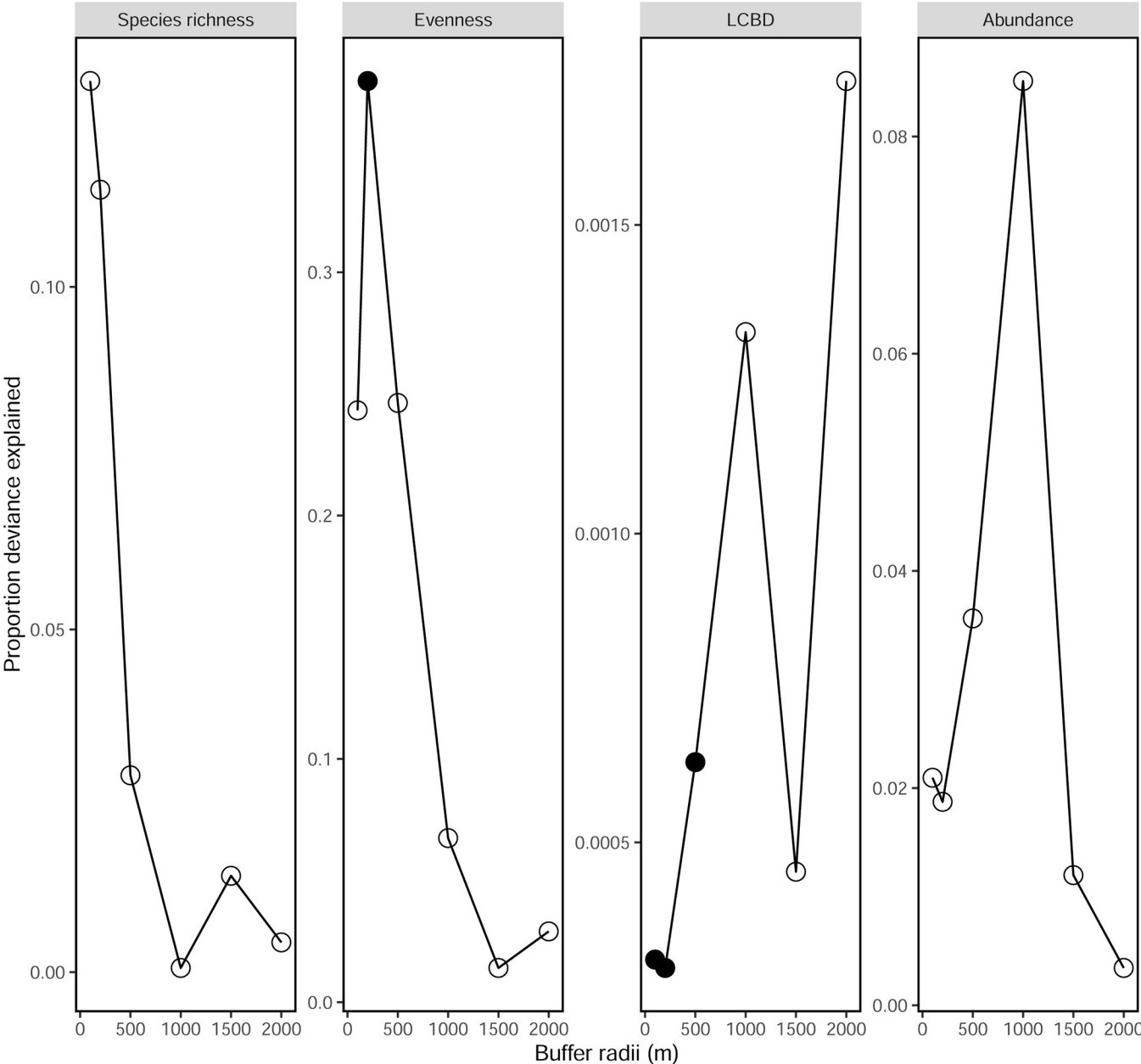

**Fig 2. Multi-scale analysis showing the proportion of deviance explained by the different estimates of parasitoid diversity and abundance when using different buffer radii (i.e. 100m, 200m, 500m, 1000m, 1500m and 2000m of radius).** Solid dots represent variables in a given buffer size with a significant effect in the mixed effects models after correction for multiple comparisons using the false discovery rate approach.

LCBD index (Fig 2). For both species richness and evenness (i.e. the Simpson diversity index), buffer sizes smaller than 500 metres explained the largest deviance, for the LCBD index the largest deviance was explained by both 1000 metres and 2000 metres buffers, even if differences among the different buffer sizes were small. For total insect abundance the largest deviance was explained by the 1000 metres buffer. When comparing models with or without a given landscape variable, landscape information was only significant at α<0.05 when extracting landscape information at 200 metres radius for models build for evenness, and when extracting data from buffers of 100, 200 and 500 metres radii for models built with the LCBD

metric. The time since agricultural practices were abandoned in each site correlated positively with the proportion of semi-natural landscape when using data from radii smaller than 200 metres, but not when using larger ones (n = 8; 100m: $S$ = 27.65, $p$-value = 0.0265; 200m: $S$ = 27.65, $p$-value = 0.0686; 500m: $S$ = 30.39, $p$-value = 0.0886; 1000m: $S$ = 46.89, $p$-value = 0.2731; 1500m: $S$ = 55.31, $p$-value = 0.4076; 2000m: $S$ = 59.47, $p$-value = 0.4816). Based on these results the 500 metres buffer was selected because this radius explained a large amount of variance in the multi-scale analysis while minimising the relationship with the time since sites were abandoned from agriculture. In addition, since some sites are located a few kilometres apart, the use of such a small radius reduces the potential for spatial autocorrelation. Similar analyses were performed using landscape information from smaller radii, and results remained very similar (S2 Table). Landscape measures and kilometric distances were not correlated as revealed by Mantel tests revealing no spatial autocorrelation among landscape-related variables (100m radius: $r$ = -0.12, $p$-value = 0.5664; 200m: $r$ = -0.16, $p$-value = 0.6744; 500m: $r$ = -0.24, $p$-value = 0.9103; 1000m: $r$ = -0.27, $p$-value = 0.9614; 1500m: $r$ = -0.21, $p$-value = 0.8385; 2000m: $r$ = -0.22, $p$-value = 0.7701).

### 3.2. Parasitoid taxonomic diversity and abundance

The variables considered did not have a significant effect on ichneumonid species richness (agricultural abandonment: est = -0.003 [-0.032, 0.025], $\chi^2_{1,6}$ = 0.08, $p$-value = 0.7795; proportion of semi-natural landscape: est = -1.407 [-5.689, 2.876], $\chi^2_{1,6}$ = 0.41, $p$-value = 0.5226; variance explained = 0.140). Species evenness (expressed as the Simpson's diversity index), related negatively with the proportion of semi-natural habitats surrounding sampling sites (est = -4.285 [-8.196, -0.375], $\chi^2_{1,6}$ = 4.44, $p$-value = 0.0351, Fig 3A), but not with the time since sites were abandoned from agriculture (est = 0.004 [-0.013, 0.028], $\chi^2_{1,6}$ = 0.18, $p$-value = 0.6711; variance explained = 0.281). Insect abundance was not affected by any of the variables considered (agricultural abandonment: est = 0.005 [-0.020, 0.031], $\chi^2_{1,6}$ = 0.25, $p$-value = 0.6166; proportion of semi-natural habitats: est = -1.553 [-5.795, 2.689], $\chi^2_{1,6}$ = 0.51, $p$-value = 0.4763; variance explained = 0.155). Beta diversity (expressed as the LCBD index) was negatively related with the proportion of semi-natural habitats at the landscape level (est = -0.072 [-1.051, -0.039], $\chi^2_{1,6}$ = 13.19, $p$-value = 0.0003, Fig 3B), but it was not affected by the time since sites were abandoned from agriculture (est = 0.001 [-0.001, 0.001], $\chi^2_{1,6}$ = 1.62, $p$-value = 0.2026; variance explained = 0.711). After correction for multiple testing none of the LCBD indices were significantly different from the others. If not corrected for multiple testing, however, three LCBD values were significant (P<0.05) and included two cropland sites in 2001 (one in Carrasqueta and one in Mariola) and the intact forest in Mariola in 2002. The mechanism behind beta diversity differences was mostly taxa replacement, which accounted for 62% (0.23), while richness differences accounted for 38% (0.14).

### 3.3. Functional diversity and abundance of the different parasitoid guilds

Functional guild richness did not relate significantly with either the time since sites were abandoned from agriculture (est = -0.004 [-0.026, 0.017], $\chi^2_{1,6}$ = 0.16, $p$-value = 0.6898), or the proportion of semi-natural habitats surrounding study sites (est = -4.516 [-10.905, 1.882], $\chi^2_{1,6}$ = 2.60, $p$-value = 0.1065; variance explained = 0.490). Similar results were found for functional guild evenness (agricultural abandonment: est = -0.001 [-0.032, 0.031], $\chi^2_{1,6}$ = 0.01, $p$-value = 0.9641; proportion of semi-natural habitats: est = 0.908 [-2.378, 4.194], $\chi^2_{1,6}$ = 0.28, $p$-value = 0.5947; variance explained = 0.436).

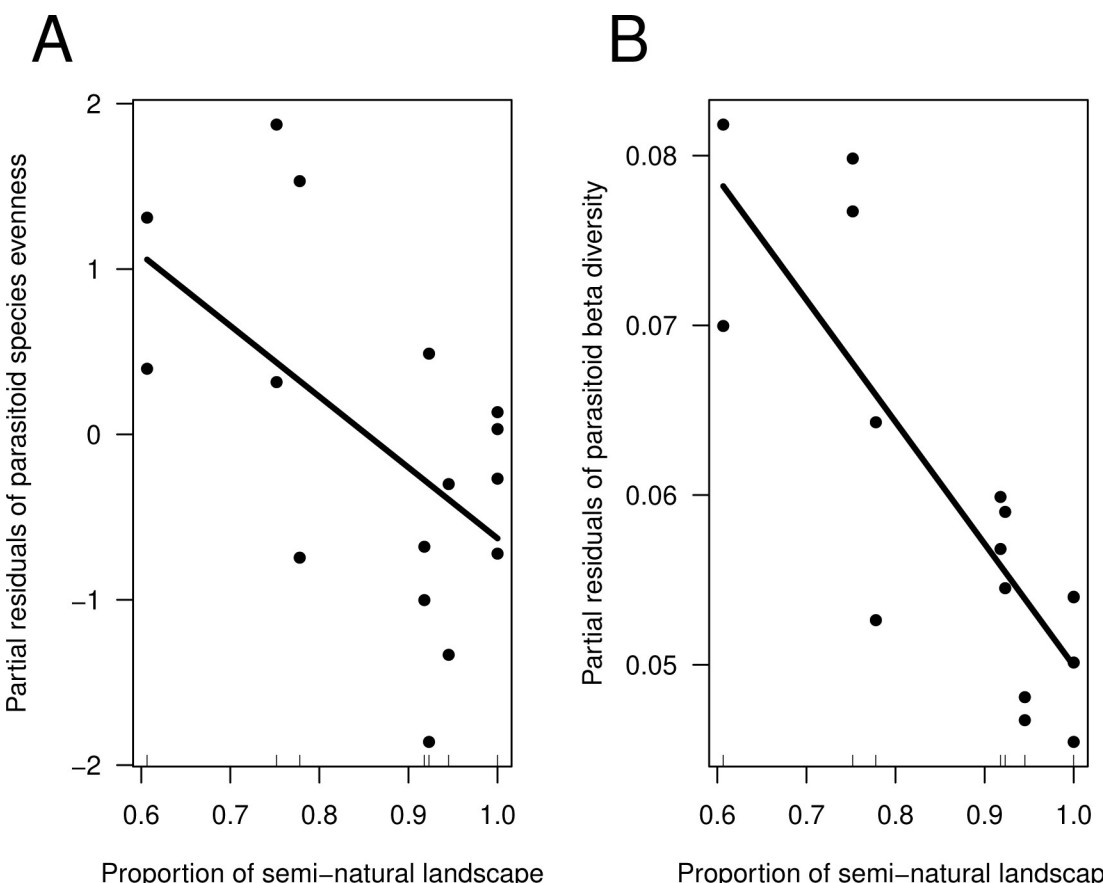

**Fig 3.** Effect of the proportion of semi-natural habitats at the landscape level on parasitoid species evenness (A) and beta diversity expressed as the LCBD index (B). Only significant effects are plotted. The fitted lines are estimated from the linear effects models, and the points represent model partial residuals based on the visreg package in R. The rugplot shows the distribution of data points along the abscissa axis.

## 4. Discussion

In this study parasitic wasps belonging to the Ichneumonidae family were collected and their diversity studied in different habitat types that varied at the local level in the time since sites were abandoned from agriculture, and at the landscape level in the proportion of semi-natural habitats. We found that alpha diversity metrics (both specific and functional) and insect abundance were not significantly affected by the time since sites were abandoned from agriculture. Contrary to what we hypothesised, the proportion of semi-natural habitat in the area surrounding study sites, negatively related with parasitoid species evenness. One potential explanation for this result may be the large percentage of semi-natural habitats in our study sites, which was always larger than 60% and that may have masked any potential effect occurring at lower percentages. This effect can also be caused by areas with a low proportion of semi-natural habitats having larger land cover diversity [68] as they may comprise, in addition to forests, semi-natural grasslands and active agricultural fields. This may have increased habitat diversity, a landscape variable known to have a strong impact on the diversity of plants and animals (e.g. [50, 69]). Semi-natural habitats and agricultural abandonment did not affect the richness and evenness of parasitoid functional guilds. These results were surprising because we expected that different habitats would host parasitoid communities that diverged at the

ecological level. Since natural enemy functional diversity has been found to be the strongest predictor of biocontrol services [34], it would be interesting to explore whether these non significant effects imply weak habitat effects on biocontrol too.

Our study was performed at eight different sites only, a limitation that may have reduced the statistical power to observe some potentially significant effects. In a recent meta-analysis Karp et al. (2018) [19] suggest that this lack of effect may also be explained by the great variation of effects found when exploring the consequences of semi-natural habitats on natural enemies. This review encompasses a total of 132 studies and reveals that non-crop habitats, for instance, can either have a positive, negative or neutral effect on natural enemy abundance and diversity. This review also reveals that the predictive power of landscape effects on natural enemy diversity can be increased when comparing similar habitats or crop plants. In hot-summer Mediterranean climates, knowledge on the impact of land-use intensification on parasitoid diversity is limited, and more studies like the one presented here are needed to unveil whether the effects found represent a common trend in this region.

Overall, our analyses revealed that relative to the agricultural practices at the sampling site, landscape-level effects had a stronger effect on parasitoid diversity. Parasitoid Hymenoptera are one of the most diverse groups of animals in terrestrial ecosystems [70]. Relative to predators, parasitoids have restricted host ranges, which makes this taxon highly sensitive to changes at lower trophic levels. These changes may include alterations in communities of insect herbivores or plants, but also in environmental conditions or habitat features, as we show here. These bottom-up cascading effects, together with the outstanding parasitoid diversity, make these organisms not only good indicators of biocontrol services, but also of environmental conditions [25, 42–44]. In a previous study Mazón and Bordera (2014) [43] studied parasitoid Hymenoptera in a protected area in central Spain, and found that diversity greatly depended on habitat type. Relative to the study presented here, parasitoid communities in this previous work were 12% more diverse than those in the current study [43]. This small difference suggests that habitats dominated by croplands interspersed with semi-natural habitats can host a large diversity of natural enemies. Our results thus add to the growing body of literature that suggests that agricultural landscapes, particularly when agricultural practices enhance plant diversity, are key to preserving diversity (e.g. [71]). Parasitoids are not often a priority in conservation programs, but their diversity, the ecosystem services they provide and their potential as indicators of environmental changes makes this group of animals potential candidates in such programs.

We found that beta diversity, expressed as local contributions to beta diversity (LCBD), decreased with an increase in the proportion of semi-natural habitat in the landscape. This change was observed even without any net gain in species richness or insect abundance, a result that reveals that important diversity patterns can only be appreciated through the lens of beta diversity [28]. Similar results have been found elsewhere, for example a study in Mexico compared communities of lepidopteran larvae among intact forests and sites in which agriculture was abandoned for 16 years. In this work, alpha diversity was not significantly different among sites, but strong differences in community composition were found [72]. Partitioning of beta diversity into species replacement and abundance components revealed that in our study the mechanism behind beta diversity differences was chiefly taxa replacement. This suggests that changes in parasitoid communities are due to different species appearing at different sites, instead of being lost. Parasitoid communities in the different sites studied are therefore quite independent of one another, and communities in croplands are likely to be independent on the migration of parasitoids from semi-natural habitats. This is possible even if several parasitoid species were shared among habitats. Spillover of natural enemies among semi-natural habitats and croplands has been largely documented [22, 24, 73], and formal tests on parasitoid

dispersal would be needed to experimentally test which parasitoid species move between habitats. Such experiments would be useful to measure the proportion of biocontrol services that are performed in croplands through parasitoid spillover from semi-natural habitats. The fact that beta diversity was larger when the proportion of semi-natural habitats was lower was unexpected because croplands are often poor at the structural level, and we did not expect to find a larger proportion of unique species in habitats dominated by agriculture. It is possible, however, that even if croplands are structurally simple, the parasitoid assemblages found are specialists of arthropods (pests or not) that can only be found in these habitats. It is also possible that parasitoids in areas dominated by croplands are unique in terms of their resilience to land-use intensification or human disturbance. In addition, the review by Tscharntke et al. (2016) [20] suggests that croplands may "provide more important resources for natural enemies than do natural habitats", which may lead to community differentiation. These authors also suggest that this may be particularly common in Mediterranean climates in which croplands have larger productivity than semi-natural habitats because they are often irrigated and artificially fertilised. For example, we found that the spider parasitoid *Trychosis legator* (Thunberg 1822) was the most abundant species both in active croplands and in those abandoned after 10 years. This may likely represent large spider diversity and abundances, which would agree with Clough et al. (2005) [74] who found that this group was highly diverse in cereal croplands. Similar results have been found elsewhere as larger diversity and abundance of natural enemies due to cropland proximity has been reported, for instance, in aphid natural enemies both in the USA [75] and in Australia [76].

To conclude, we see in our study that parasitoids were very diverse and therefore likely to act as important organisms in the agricultural ecosystems studied through suppression of herbivorous arthropods. Natural enemy diversity often correlates with biocontrol services, but in diverse communities antagonistic interactions like intraguild predation or hyperparasitism may emerge [23]. These interactions may have negative consequences for pest biocontrol as has often been documented [15]. In the current study, beta diversity correlated with the proportion of semi-natural habitats in the landscape. Even if we did not measure biocontrol services, doing so would be very interesting to test the effect that different types of semi-natural habitats, with different beta diversity values, may have on biocontrol services. In the study area, croplands are often of small size, and semi-natural habitats are always found nearby. Mediterranean habitats are at great risk due to global changes and habitat loss [77], a situation that can be mitigated through policies that promote the presence of semi-natural habitats with the potential to suppress pest outbreaks [8]. One good way to implement such policies is not only by promoting such semi-natural habitats, but also to find the right landscape structures and agricultural practices that facilitate the spillover of natural enemies from semi-natural habitats to croplands [71]. The success of these techniques, however, may require assessing the scale at which diversity patterns operate along landscape gradients, and whether the beneficial parasitoid assemblages that originate from semi-natural habitats are maintained in large croplands.

## Supporting information

**S1 Table. Absolute numbers of individuals of all subfamilies caught in each habitat, with data of the trophic guild and biological strategy where they were classified into.** Habitats: MasC: Mas de Cano, MasI: Mas de San Ignacio, Men: Menejador, VenC: VCarrasqueta, CavB: Caveta del Buitre, FoA: Foia Ampla, MasP: Mas del Parral, Ret: Retura; Trophic guilds: Coc: parasitoids of cocoons, Fun: parasitoid of fungiphage larvae, cPh: parasitoid of concealed phytophages, gPh: grazing phytophages, Mel: parasitoid of melitophages, Pol: poliphage

parasitoids, Sap: parasitoid of saprophages, Xyl: parasitoid of xylophages, Unk: hosts unknown, Zoo: parasitoid of zoophages. Strategy: K: koinobionts, I: idiobionts Ec: ectoparasitoids, En: endoparasitoids.
(DOCX)

**S2 Table. Results of statistical tests using various radii.**
(DOCX)

## Acknowledgments

We want to express our gratitude to the staff of Sierra Mariola Natural Park and Font Roja Natural Park for providing us facilities and permissions for collecting material in these protected areas.

## Author Contributions

**Conceptualization:** Marina Mazón, Santiago Bordera, Enric Frago.

**Data curation:** Marina Mazón, Alexander Rodríguez-Berrío, Enric Frago.

**Formal analysis:** Marina Mazón, Enric Frago.

**Funding acquisition:** Marina Mazón, Santiago Bordera.

**Investigation:** Marina Mazón, Santiago Bordera, Alexander Rodríguez-Berrío, Enric Frago.

**Methodology:** Marina Mazón, Santiago Bordera, Alexander Rodríguez-Berrío, Enric Frago.

**Project administration:** Santiago Bordera.

**Resources:** Santiago Bordera.

**Validation:** Alexander Rodríguez-Berrío.

**Writing – original draft:** Marina Mazón, Enric Frago.

**Writing – review & editing:** Marina Mazón, Santiago Bordera, Enric Frago.

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
