## [Decision Letter · Decision Letter 0]

6 Oct 2023

PONE-D-23-25676The relative influence of agricultural abandonment and semi-natural habitats on parasitoid diversity and community compositionPLOS ONE

Dear Dr. Frago,

Thank you for submitting your manuscript to PLOS ONE. After careful consideration, we feel that it has merit but does not fully meet PLOS ONE’s publication criteria as it currently stands. Therefore, we invite you to submit a revised version of the manuscript that addresses the points raised during the review process.

We look forward to receiving your revised manuscript.

Kind regards,

Patrick R Stephens, Ph.D.

Academic Editor

PLOS ONE

Journal Requirements:

 "This study was supported by the Projects BOS2000-0148 from Ministero de Ciencia y Tecnología (D.G.I.) of Spanish Government and GV06/271 from Conselleria d’Empresa, Universtitat i Ciència of Generalitat Valenciana (Spain). AR-B was financed by a PhD research grant MAE-AECI (2003-2006) of Agencia Española de Cooperación Internacional (Spanish Goverment). EF is currently funded by the Agence Nationale de la Recherche (ANR) via the ANR ENEMYCOCKTAIL project and by CIRAD"

"We want to express our gratitude to the staff of Sierra Mariola Natural Park and Font Roja

Natural Park for providing us facilities and permissions for collecting material in these

protected areas. This study was supported by the Projects BOS2000-0148 from Ministero de

Ciencia y Tecnología (D.G.I.) of Spanish Government and GV06/271 from Conselleria

d’Empresa, Universtitat i Ciència of Generalitat Valenciana (Spain). AR-B was financed by a

PhD research grant MAE-AECI (2003-2006) of Agencia Española de Cooperación

Internacional (Spanish Goverment). EF is currently funded by the Agence Nationale de la

Recherche (ANR) via the ANR ENEMYCOCKTAIL project and by CIRAD."

 "This study was supported by the Projects BOS2000-0148 from Ministero de Ciencia y Tecnología (D.G.I.) of Spanish Government and GV06/271 from Conselleria d’Empresa, Universtitat i Ciència of Generalitat Valenciana (Spain). AR-B was financed by a PhD research grant MAE-AECI (2003-2006) of Agencia Española de Cooperación Internacional (Spanish Goverment). EF is currently funded by the Agence Nationale de la Recherche (ANR) via the ANR ENEMYCOCKTAIL project and by CIRAD"

6. Ethics statement appears in the Methods section of the manuscript AND at the end of the manuscript:

Your ethics statement should only appear in the Methods section of your manuscript. If your ethics statement is written in any section besides the Methods, please delete it from any other section.

7. Please include a separate caption for each figure in your manuscript.

8. lease include your tables as part of your main manuscript and remove the individual files. Please note that supplementary tables (should remain/ be uploaded) as separate ""supporting information"" files"""

**Additional Editor Comments:**

Both reviewers found the methods generally sound, but also suggested specific revisions that are needed before it will be suitable for publication.  In particular, both reviewers felt that additional clarification concerning the hypotheses of interest is needed.  I hope that you find these comments helpful as a basis of a revision.

Reviewers' comments:

Reviewer's Responses to Questions

**Comments to the Author**

1. Is the manuscript technically sound, and do the data support the conclusions?

Reviewer #1: Partly

Reviewer #2: No

2. Has the statistical analysis been performed appropriately and rigorously? 

Reviewer #1: No

Reviewer #2: No

3. Have the authors made all data underlying the findings in their manuscript fully available?

Reviewer #1: Yes

Reviewer #2: No

4. Is the manuscript presented in an intelligible fashion and written in standard English?

Reviewer #1: Yes

Reviewer #2: Yes

5. Review Comments to the Author

Reviewer #1: The paper addresses an important subject about the influence of the time of agricultural abandonment and proportion of semi-natural habitats on taxonomic and functional diversity of parasitoids. It is an interesting topic and the text is clear and easy to follow. My main concerns are the number of sampled sites that are very few and some issues in introduction and discussion. Particularly, the mechanisms for which time of agricultural abandonment (local scale) could affect parasitoids are not explicitly explained. For example, could one driver be the increasing plant richness and biomass with increasing time of agricultural abandonment? I understand the authors did not measure any of the variables related to plants and herbivores, but I consider that more explanations of how lower trophic levels can change with the time of abandonment are needed. Another important point is about this explanatory variable which is considered and analyzed by the authors as a continuous variable of different times of abandonment, however it involves only a few levels of time: not abandoned (active croplands), recently abandoned (10 and 20 years ago abandoned crops) and wild sites (considered as 75 years ago abandoned crops) and these look like different stages of succession of sites.

Specific points

Introduction

- 4th paragraph. “the relative contribution of two different scales to alpha and beta”, do authors mean spatial scales?

- Last paragraph. Points 1 to 4 are confusing because I do not understand if they are a list of the hypothesis or predictions and/or questions to be answered.

Methods

- Paragraph of study area. It would be useful to mention here the most dominant plant species in semi-natural habitats.

- It is not clear the explanation about selection and number of sites. It seems that authors sampled 12 sites in total (6 in each mountain) according to the written “In each mountain range, we selected four sites along a gradient of time since agriculture was abandoned, from a wild protected area to a managed cropland, and two sites at different stages of succession (i.e. ten and twenty years from agricultural abandonment). And also, it is confusing why only two sites are in different stages of succession.

Reviewer #2: This is a very well written paper on the variation in diversity of parasitoid wasps amongst landscape composition. Unfortunately, I believe there are several caveats to this study that prevent the authors from achieving their goals and answering their scientific questions.

First, the authors start with a very nice and throughout background section in their introduction, which is sadly followed by unclear hypotheses statements. For instance, just the first hypothesis could be partitioned into 4 different ones! Even more, the 2nd and 4th hypothesis seem to mix literature background that has not been introduced previously, making them quite hard to fully understand at this stage of the paper! I would suggest the authors to make a comprehensive figure, with graphs showing the expected directionality of each hypothesis, I believe it would make the text much clearer that way.

Second, and this is perhaps my biggest comment, the number of sites that the authors have chosen is not enough to test the hypotheses they put forward. In table 1, the authors present a brief description of their site, in which we can see that there are only 2 replicates per group. The authors briefly acknowledge this is the discussion, but, to me, it severely hinders their capacity to make successful statistical analysis! The major problem in here is that the sites are not spatially isolated (15km of distance between the 2 sites), which means that the sites could be considered as pseudo-replicates at the spatial extent at which the authors work. For instance, considering the larger buffer size at 2000m around the sites for each site would leave only 11km between the 2!

Also, as a more minor comment, the statistical analysis section in the method is not clear and needs to be reworded. There are a lot of information in there that are mixed with other information, such as the last sentence of the section. There is also some information missing in the model building, such as Distribution family for each response variables. Please provide the data as inputted in R + R code to make it clear for the reader.

The result section also needs a lot of work and rewording. I do not understand the S value in the result section. Does it refer to the effect size of your models? If so, please provide 95% CI for each predictor. P-Values do not present any useful information (Table 2). As I stated above, please present Effect size of each predictor + 95% CI instead.

Minor comments:

Methods:

I do not understand why you gave numerical values to your sites, and it is not clear how you performed the analysis.

Make your pace of sampling clearer in the methods

6. PLOS authors have the option to publish the peer review history of their article (what does this mean?). If published, this will include your full peer review and any attached files.

Reviewer #1: No

Reviewer #2: No

---

## [Decision Letter · Decision Letter 1]

18 Mar 2024

PONE-D-23-25676R1The relative influence of agricultural abandonment and semi-natural habitats on parasitoid diversity and community compositionPLOS ONE

Dear Dr. Frago,

Thank you for submitting your manuscript to PLOS ONE. After careful consideration, we feel that it has merit but does not fully meet PLOS ONE’s publication criteria as it currently stands. Therefore, we invite you to submit a revised version of the manuscript that addresses the points raised during the review process.

Both reviewers found the manuscript much improved.  However, a few minor issues still need to be addressed.  My thanks for your efforts. Please submit your revised manuscript by May 02 2024 11:59PM. If you will need more time than this to complete your revisions, please reply to this message or contact the journal office at plosone@plos.org. Please include the following items when submitting your revised manuscript:A rebuttal letter that responds to each point raised by the academic editor and reviewer(s). You should upload this letter as a separate file labeled 'Response to Reviewers'.A marked-up copy of your manuscript that highlights changes made to the original version. You should upload this as a separate file labeled 'Revised Manuscript with Track Changes'.An unmarked version of your revised paper without tracked changes. You should upload this as a separate file labeled 'Manuscript'.If applicable, we recommend that you deposit your laboratory protocols in protocols.io to enhance the reproducibility of your results. Protocols.io assigns your protocol its own identifier (DOI) so that it can be cited independently in the future. For instructions see: https://journals.plos.org/plosone/s/submission-guidelines#loc-laboratory-protocols. Additionally, PLOS ONE offers an option for publishing peer-reviewed Lab Protocol articles, which describe protocols hosted on protocols.io. Read more information on sharing protocols at https://plos.org/protocols?utm_medium=editorial-email&utm_source=authorletters&utm_campaign=protocols.

We look forward to receiving your revised manuscript.

Kind regards,

Patrick R Stephens, Ph.D.

Academic Editor

PLOS ONE

Journal Requirements:

Reviewers' comments:

Reviewer's Responses to Questions

**Comments to the Author**

1. If the authors have adequately addressed your comments raised in a previous round of review and you feel that this manuscript is now acceptable for publication, you may indicate that here to bypass the “Comments to the Author” section, enter your conflict of interest statement in the “Confidential to Editor” section, and submit your "Accept" recommendation.

Reviewer #2: (No Response)

Reviewer #3: (No Response)

2. Is the manuscript technically sound, and do the data support the conclusions?

Reviewer #2: Yes

Reviewer #3: Yes

3. Has the statistical analysis been performed appropriately and rigorously? 

Reviewer #2: Yes

Reviewer #3: Yes

4. Have the authors made all data underlying the findings in their manuscript fully available?

Reviewer #2: Yes

Reviewer #3: Yes

5. Is the manuscript presented in an intelligible fashion and written in standard English?

Reviewer #2: Yes

Reviewer #3: Yes

6. Review Comments to the Author

Reviewer #2: The authors have addressed most of my comments. However, there is still a bit of work to do, at least in my opinion.

L280: A gaussian distribution is bound by -inf to +inf for continuous variables, and therefore doesn't suit any of the investigated response variable. I understand this sounds trivial, and will probably not affect the results very much. It is however, a flawed assumption to choose the gaussian as default in mixed modelling. You can refer to books such as Zuur et al. 2009 for more appropriate distribution families for the models you are running.

It is also my opinion that p-values poorly reflect the range of effect size from the models. For instance, we could have a scenario in which other researchers would use the effect size you present for a meta-analysis. Without a CI range, it becomes impossible to fully understand the potential range for errors, as p-values only reflects if said range crosses the 0 line or not. As such, I would strongly recommend the authors to add the CI in their results, and keep the p-values if they want to do so.

Minor comments:

L244: it's 'kilometers', not 'quilometers'

Aside from that, I'm quite happy with the manuscript in its present form, and congratulate the authors for a thorough work in addressing my comments!

Bon travail!

Antoine Filion, PhD

Reviewer #3: Comments to authors

This paper is described parasitoid wasp diversity in agricultural abandonment and semi-natural habitats and provide insight to relative influence of different habitats for them selecting one wasp family as a sample organism.

Comment 1:

“In this study (1) we hypothesise that parasitoid species richness, evenness, total abundance and beta diversity will relate positively with the time since agriculture was abandoned, and with the proportion of semi-natural habitats at the landscape level.”

Here, could you please explain what is meant by “with the proportion of semi-natural habitats at the landscape level”?

Comment 2:

It is very clear that authors have selected four sites in each mountain range. But it is bit unclear in the following sentence “ and two sites at different stages of succession (i.e. ten and twenty years from agricultural abandonment)”.

“In each mountain range, we selected four sites along a gradient of time since agriculture was abandoned, from a wild protected area to a managed cropland, and two sites at different stages of succession (i.e. ten and twenty years from agricultural abandonment)”

Comment 3:

It would be better that scientific name of the species followed by author & year (year they described the species) eg: Mesostenus albinotatus Gravenhorst, YEAR

Comment 4:

Appendix 1: It would be convenient for readers if habitats are classified as cropland, wild etc.. rather than listing their names.

7. PLOS authors have the option to publish the peer review history of their article (what does this mean?). If published, this will include your full peer review and any attached files.

Reviewer #2: **Yes: **Antoine Filion, PhD

Reviewer #3: **Yes: **Sasanka Ranasinghe

---

## [Author Response · Author response to Decision Letter 1]

10 Apr 2024

See reply to reviewers letter attached

---

## [Decision Letter · Decision Letter 2]

30 Apr 2024

The relative influence of agricultural abandonment and semi-natural habitats on parasitoid diversity and community composition

PONE-D-23-25676R2

Dear Dr. Frago,

We’re pleased to inform you that your manuscript has been judged scientifically suitable for publication and will be formally accepted for publication once it meets all outstanding technical requirements.

Kind regards,

Patrick R Stephens, Ph.D.

Academic Editor

PLOS ONE

Additional Editor Comments (optional):

Reviewers' comments:

Reviewer's Responses to Questions

**Comments to the Author**

1. If the authors have adequately addressed your comments raised in a previous round of review and you feel that this manuscript is now acceptable for publication, you may indicate that here to bypass the “Comments to the Author” section, enter your conflict of interest statement in the “Confidential to Editor” section, and submit your "Accept" recommendation.

Reviewer #2: All comments have been addressed

Reviewer #3: All comments have been addressed

2. Is the manuscript technically sound, and do the data support the conclusions?

Reviewer #2: Yes

Reviewer #3: Yes

3. Has the statistical analysis been performed appropriately and rigorously? 

Reviewer #2: Yes

Reviewer #3: Yes

4. Have the authors made all data underlying the findings in their manuscript fully available?

Reviewer #2: Yes

Reviewer #3: Yes

5. Is the manuscript presented in an intelligible fashion and written in standard English?

Reviewer #2: Yes

Reviewer #3: Yes

6. Review Comments to the Author

Reviewer #2: (No Response)

Reviewer #3: Authors have addresses my previous comments in an understandable way. They are included in the manuscript. good!

7. PLOS authors have the option to publish the peer review history of their article (what does this mean?). If published, this will include your full peer review and any attached files.

Reviewer #2: **Yes: **Antoine Filion

Reviewer #3: **Yes: **Sasanka Ranasinghe

---

## [Editor Report · Acceptance letter]

23 May 2024

PONE-D-23-25676R2 

PLOS ONE

Dear Dr. Frago, 

I'm pleased to inform you that your manuscript has been deemed suitable for publication in PLOS ONE. Congratulations! Your manuscript is now being handed over to our production team.

Kind regards, 

on behalf of

Dr. Patrick R Stephens 

Academic Editor

PLOS ONE